# Bayesian Deep Learning and a Probabilistic Perspective of Generalization

**Andrew Gordon Wilson**
New York University

**Pavel Izmailov**
New York University

## Abstract

The key distinguishing property of a Bayesian approach is marginalization, rather than using a single setting of weights. Bayesian marginalization can particularly improve the accuracy and calibration of modern deep neural networks, which are typically underspecified by the data, and can represent many compelling but different solutions. We show that deep ensembles provide an effective mechanism for approximate Bayesian marginalization, and propose a related approach that further improves the predictive distribution by marginalizing within basins of attraction, without significant overhead. We also investigate the prior over functions implied by a vague distribution over neural network weights, explaining the generalization properties of such models from a probabilistic perspective. From this perspective, we explain results that have been presented as mysterious and distinct to neural network generalization, such as the ability to fit images with random labels, and show that these results can be reproduced with Gaussian processes. We also show that Bayesian model averaging alleviates double descent, resulting in monotonic performance improvements with increased flexibility.

## 1 Introduction

Imagine fitting the airline passenger data in Figure 1. Which model would you choose: (1) $f_1(x) = w_0 + w_1 x$, (2) $f_2(x) = \sum_{j=0}^{3} w_j x^j$, or (3) $f_3(x) = \sum_{j=0}^{10^4} w_j x^j$?

Put this way, most audiences overwhelmingly favour choices (1) and (2), for fear of overfitting. But of these options, choice (3) most honestly represents our beliefs. Indeed, it is likely that the ground truth explanation for the data is out of class for any of these choices, but there is some setting of the coefficients $\{w_j\}$ in choice (3) which provides a better description of reality than could be managed by choices (1) and (2), which are special cases of choice (3). Moreover, our beliefs about the generative processes for our observations, which are often very sophisticated, typically ought to be independent of how many data points we observe.

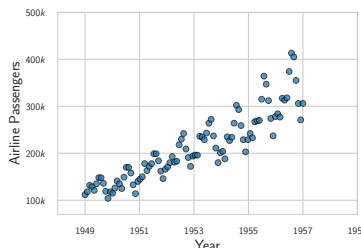

Figure 1: Airline passenger data.

And in modern practice, we are implicitly favouring choice (3): we often use neural networks with millions of parameters to fit datasets with thousands of points. Furthermore, non-parametric methods such as Gaussian processes often involve infinitely many parameters, enabling the flexibility for universal approximation [40], yet in many cases provide very simple predictive distributions. Indeed, parameter counting is a poor proxy for understanding generalization behaviour.

From a probabilistic perspective, we argue that generalization depends largely on *two* properties, the *support* and the *inductive biases* of a model. Consider Figure 2(a), where on the horizontal axis we have a conceptualization of all possible datasets, and on the vertical axis the Bayesian *evidence* for a

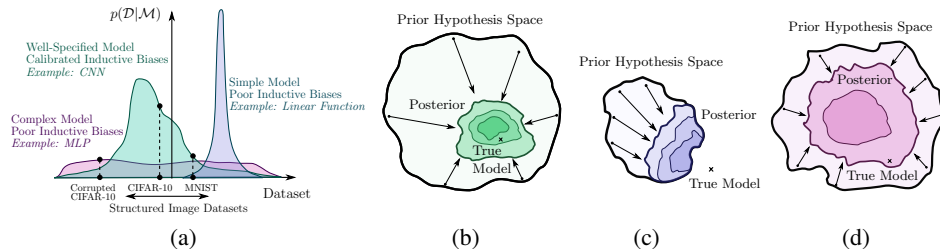

Figure 2: **A probabilistic perspective of generalization.** (a) Ideally, a model supports a wide range of datasets, but with inductive biases that provide high prior probability to a particular class of problems being considered. Here, the CNN is preferred over the linear model and the fully-connected MLP for CIFAR-10 (while we do not consider MLP models to in general have poor inductive biases, here we are considering a hypothetical example involving images and a very large MLP). (b) By representing a large hypothesis space, a model can contract around a true solution, which in the real-world is often very sophisticated. (c) With truncated support, a model will converge to an erroneous solution. (d) Even if the hypothesis space contains the truth, a model will not efficiently contract unless it also has reasonable inductive biases.

model. The evidence, or marginal likelihood, $p(\mathcal{D}|\mathcal{M}) = \int p(\mathcal{D}|\mathcal{M}, w)p(w)dw$, is the probability we would generate a dataset if we were to randomly sample from the prior over functions $p(f(x))$ induced by a prior over parameters $p(w)$. We define the support as the range of datasets for which $p(\mathcal{D}|\mathcal{M}) > 0$. We define the inductive biases as the relative prior probabilities of different datasets — the *distribution of support* given by $p(\mathcal{D}|\mathcal{M})$. A similar schematic to Figure 2(a) was used by MacKay [26] to understand an Occam's razor effect in using the evidence for model selection; we believe it can also be used to reason about model construction and generalization.

From this perspective, we want the support of the model to be large so that we can represent any hypothesis we believe to be possible, even if it is unlikely. We would even want the model to be able to represent pure noise, such as noisy CIFAR [51], as long as we honestly believe there is some non-zero, but potentially arbitrarily small, probability that the data are simply noise. Crucially, we also need the inductive biases to carefully represent which hypotheses we believe to be a priori likely for a particular problem class. If we are modelling images, then our model should have statistical properties, such as convolutional structure, which are good descriptions of images.

Figure 2(a) illustrates three models. We can imagine the blue curve as a simple linear function, $f(x) = w_0 + w_1 x$, combined with a distribution over parameters $p(w_0, w_1)$, e.g., $\mathcal{N}(0, I)$, which induces a distribution over functions $p(f(x))$. Parameters we sample from our prior $p(w_0, w_1)$ give rise to functions $f(x)$ that correspond to straight lines with different slopes and intercepts. This model thus has truncated support: it cannot even represent a quadratic function. But because the marginal likelihood must normalize over datasets $\mathcal{D}$, this model assigns much mass to the datasets it does support. The red curve could represent a large fully-connected MLP. This model is highly flexible, but distributes its support across datasets too evenly to be particularly compelling for many image datasets. The green curve could represent a convolutional neural network, which represents a compelling specification of support and inductive biases for image recognition: this model is highly flexible, but it provides a particularly good support for structured problems.

With large support, we cast a wide enough net that the posterior can contract around the true solution to a given problem as in Figure 2(b), which in reality we often believe to be very sophisticated. On the other hand, the simple model will have a posterior that contracts around an erroneous solution if it is not contained in the hypothesis space as in Figure 2(c). Moreover, in Figure 2(d), the model has wide support, but does not contract around a good solution because its support is too evenly distributed.

Returning to the opening example, we can justify the high order polynomial by wanting large support. But we would still have to carefully choose the prior on the coefficients to induce a distribution over functions that would have reasonable inductive biases. Indeed, this Bayesian notion of generalization is not based on a single number, but is a two dimensional concept. From this probabilistic perspective, it is crucial not to conflate the *flexibility* of a model with the *complexity* of a model class. Indeed Gaussian processes with RBF kernels have large support, and are thus flexible, but have inductive

biases towards very simple solutions. We also see that *parameter counting* has no significance in this perspective of generalization: what matters is how a distribution over parameters combines with a functional form of a model, to induce a distribution over solutions.

In this paper we reason about Bayesian deep learning from a probabilistic perspective of generalization. The key distinguishing property of a Bayesian approach is marginalization instead of optimization, where we represent solutions given by all settings of parameters weighted by their posterior probabilities, rather than bet everything on a single setting of parameters. Neural networks are typically underspecified by the data, and can represent many different but high performing models corresponding to different settings of parameters, which is exactly when marginalization will make the biggest difference for accuracy and calibration. Moreover, we clarify that the recent deep ensembles [22] are not a competing approach to Bayesian inference, but can be viewed as a compelling mechanism for Bayesian marginalization. Indeed, we empirically demonstrate that deep ensembles can provide a *better* approximation to the Bayesian predictive distribution than standard Bayesian approaches. We propose MultiSWAG, a method inspired by deep ensembles, which marginalizes within basins of attraction — achieving improved performance, with a similar training time.

We then investigate the properties of priors over functions induced by priors over the weights of neural networks, showing that they have reasonable inductive biases, and connect these results to tempering. We also show that the mysterious generalization properties recently presented in Zhang et al. [51] can be understood by reasoning about prior distributions over functions, and are not specific to neural networks. Indeed, we show Gaussian processes can also perfectly fit images with random labels, yet generalize on the noise-free problem. These results are a consequence of large support but reasonable inductive biases for common problem settings. We further show that while Bayesian neural networks can fit the noisy datasets, the marginal likelihood has much better support for the noise free datasets, in line with Figure 2. We additionally show that the multimodal marginalization in MultiSWAG alleviates double descent, so as to achieve monotonic improvements in performance with model flexibility, in line with our perspective of generalization. MultiSWAG also provides significant improvements in both accuracy and NLL over SGD training and unimodal marginalization.

We provide code at `https://github.com/izmailovpavel/understandingbdl`.

## 2   Related Work

Notable early works on Bayesian neural networks include MacKay [26], MacKay [27], and Neal [35]. These works generally argue in favour of making the model class for Bayesian approaches as flexible as possible, in line with Box and Tiao [5]. Accordingly, Neal [35] pursued the limits of large Bayesian neural networks, showing that as the number of hidden units approached infinity, these models become Gaussian processes with particular kernel functions. This work harmonizes with recent work describing the neural tangent kernel [e.g., 16].

The marginal likelihood is often used for Bayesian hypothesis testing, model comparison, and hyperparameter tuning, with *Bayes factors* used to select between models [18]. MacKay [28, Ch. 28] uses a diagram similar to Fig 2(a) to show the marginal likelihood has an *Occam's razor* property, favouring the simplest model consistent with a given dataset, even if the prior assigns equal probability to the various models. Rasmussen and Ghahramani [41] reasons about how the marginal likelihood can favour large flexible models, as long as they correspond to a reasonable distribution over functions.

There has been much recent interest in developing Bayesian approaches for modern deep learning, with new challenges and architectures quite different from what had been considered in early work. Recent work has largely focused on scalable inference [e.g., 4, 9, 19, 42, 20, 29], function-space inspired priors [e.g., 50, 25, 45, 13], and developing flat objective priors in parameter space, directly leveraging the biases of the neural network functional form [e.g, 34]. Wilson [48] provides a note motivating Bayesian deep learning.

In general, PAC-Bayes provides a compelling framework for deriving explicit non-asymptotic generalization bounds [31, 23, 7, 36, 37, 30, 17]. These bounds can be improved by, e.g. fewer parameters, and very compact priors, which can be different from what provides optimal generalization. From our perspective, model flexibility and priors with *large* support, rather than compactness, are desirable. Our work also shows the importance of multi-basin marginalization for generalization in deep learning, while the PAC-Bayes bounds are essentially unchanged by a multi-modal posterior.

Our focus is complementary to PAC-Bayes, and largely prescriptive, aiming to provide intuitions on model construction, inference, generalization, and neural network priors, as well as new connections between Bayesian model averaging and deep ensembles, benefits of Bayesian model averaging specifically in the context of modern deep neural networks, perspectives on tempering in Bayesian deep learning, views of marginalization that contrast with simple Monte Carlo, and new methods for Bayesian marginalization in deep learning.

In other work, Pearce et al. [39] propose a modification of deep ensembles and argue that it performs approximate Bayesian inference, and Gustafsson et al. [12] briefly mention how deep ensembles can be viewed as samples from an approximate posterior. Fort et al. [8] considered the diversity of predictions produced by models from a single SGD run, and models from independent SGD runs, and suggested to ensemble averages of SGD iterates.

# 3 Bayesian Marginalization

Often the predictive distribution we want to compute is given by

$$p(y|x, \mathcal{D}) = \int p(y|x, w)p(w|\mathcal{D})dw \,. \tag{1}$$

The outputs are $y$ (e.g., regression values, class labels, ...), indexed by inputs $x$ (e.g. spatial locations, images, ...), the weights (or parameters) of the neural network $f(x; w)$ are $w$, and $\mathcal{D}$ are the data. Eq. (1) represents a *Bayesian model average* (BMA). Rather than bet everything on one hypothesis — with a single setting of parameters $w$ — we want to use all settings of parameters, weighted by their posterior probabilities. This procedure is called *marginalization* of the parameters $w$, as the predictive distribution of interest no longer conditions on $w$. This is not a controversial equation, but simply the sum and product rules of probability.

## 3.1 Beyond Monte Carlo

Nearly all approaches to estimating the integral in Eq. (1), when it cannot be computed in closed form, involve a *simple Monte Carlo* approximation: $p(y|x, \mathcal{D}) \approx \frac{1}{J} \sum_{j=1}^{J} p(y|x, w_j) \,, w_j \sim p(w|\mathcal{D})$. In practice, the samples from the posterior $p(w|\mathcal{D})$ are also approximate, and found through MCMC or deterministic methods. The deterministic methods approximate $p(w|\mathcal{D})$ with a different more convenient density $q(w|\mathcal{D}, \theta)$ from which we can sample, often chosen to be Gaussian. The parameters $\theta$ are selected to make $q$ close to $p$ in some sense; for example, variational approximations [e.g., 2], which have emerged as a popular deterministic approach, find $\operatorname{argmin}_\theta \mathcal{KL}(q||p)$. Other standard deterministic approximations include Laplace [e.g., 27], EP [32], and INLA [43].

From the perspective of estimating the predictive distribution in Eq. (1), we can view simple Monte Carlo as approximating the posterior with a set of point masses, with locations given by samples from another approximate posterior $q$, even if $q$ is a continuous distribution. That is, $p(w|\mathcal{D}) \approx \sum_{j=1}^{J} \delta(w = w_j) \,, w_j \sim q(w|\mathcal{D})$.

Ultimately, the goal is to accurately compute the predictive distribution in Eq. (1), rather than find a generally accurate representation of the posterior. In particular, we must carefully represent the posterior in regions that will make the greatest contributions to the BMA integral. In Sections 3.2 and 4, we consider how various approaches approximate the predictive distribution.

## 3.2 Deep Ensembles are BMA

*Deep ensembles* [22] is fast becoming a gold standard for accurate and well-calibrated predictive distributions. Recent reports [e.g., 38, 1] show that deep ensembles appear to outperform some particular approaches to Bayesian neural networks for uncertainty representation, leading to the confusion that deep ensembles and Bayesian methods are competing approaches. These methods are often explicitly referred to as non-Bayesian [e.g., 22, 38, 47]. To the contrary, we argue that deep ensembles are actually a compelling approach to BMA, in the vein of Section 3.1.

Furthermore, by representing multiple basins of attraction, deep ensembles can provide a *better* approximation to the BMA than the Bayesian approaches in Ovadia et al. [38]. Indeed, the functional

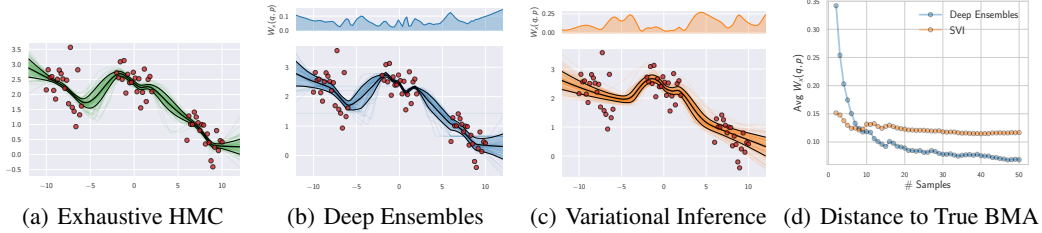

| (a) Exhaustive HMC | (b) Deep Ensembles | (c) Variational Inference | (d) Distance to True BMA |

Figure 3: **Approximating the true predictive distribution. (a)**: A close approximation of the true predictive distribution obtained by combining 10 long HMC chains. **(b)**: Deep ensembles predictive distribution using 50 independently trained networks. **(c)**: Predictive distribution for factorized variational inference (VI). **(d)**: Convergence of the predictive distributions for deep ensembles and variational inference as a function of the number of samples; we measure the average Wasserstein distance between the marginals in the range of input positions. The multi-basin deep ensembles approach provides a more faithful approximation of the Bayesian predictive distribution than the conventional single-basin VI approach, which is overconfident between data clusters. The top panels show the Wasserstein distance between the true predictive distribution and the deep ensemble and VI approximations, as a function of inputs $x$.

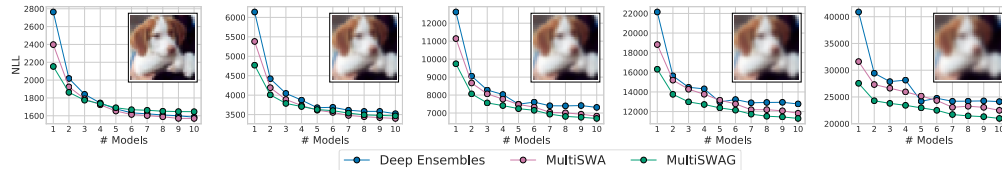

Figure 4: Negative log likelihood for Deep Ensembles, MultiSWAG and MultiSWA using a PreResNet-20 on CIFAR-10 with varying intensity of the *Gaussian blur* corruption. The image in each plot shows the intensity of corruption. For all levels of intensity, MultiSWAG and MultiSWA outperform Deep Ensembles for a small number of independent models. For high levels of corruption MultiSWAG significantly outperforms other methods even for many independent models. We present results for other corruptions in the Appendix.

diversity is important for a good approximation to the BMA integral, as per Section 3.1. We explore these questions in Section 4.

## 4    An Empirical Study of Marginalization

We have shown that deep ensembles can be interpreted as an approximate approach to Bayesian marginalization, which selects for functional diversity by representing multiple basins of attraction in the posterior. Most Bayesian deep learning methods instead focus on faithfully approximating a posterior within a single basin of attraction. We propose a new method, MultiSWAG, which combines these two types of approaches. MultiSWAG combines multiple independently trained SWAG approximations [29], to create a mixture of Gaussians approximation to the posterior, with each Gaussian centred on a different basin. We note that MultiSWAG does not require *any* additional training time over standard deep ensembles. We illustrate the conceptual difference between deep ensembles, a standard variational single basin approach, and MultiSWAG, in Figure 8 (Appendix).

In Figure 3 we evaluate single basin and multi-basin approaches in a case where we can near-exactly compute the predictive distribution. To approximate the ground truth, we use 10 chains of Hamiltonian Monte Carlo (HMC) from the `hamiltorch` package [6]. We provide details for generating the data and training the models as well as convergence analysis for our HMC sampler in Appendix D.1. We see that the predictive distribution given by deep ensembles is qualitatively closer to the true distribution, compared to the single basin variational method: between data clusters, the deep ensemble approach provides a similar representation of epistemic uncertainty to exhaustive HMC, whereas the variational method is extremely overconfident in these regions. Moreover, we see that the Wasserstein distance between the true predictive distribution and these two approximations

quickly shrinks with number of samples for deep ensembles, but is roughly independent of number of samples for the variational approach. Thus the deep ensemble is providing a better approximation of the Bayesian model average in Eq. (1) than the single basin variational approach, which has traditionally been labelled as the Bayesian alternative. The variational approach would need to marginalize over multiple basins to be competitive with deep ensembles as an approximation to the Bayesian predictive distribution.

Next, we evaluate MultiSWAG under distribution shift on the CIFAR-10 dataset [21], replicating the setup in Ovadia et al. [38]. We consider 16 data corruptions, each at 5 different levels of severity, introduced by Hendrycks and Dietterich [14]. For each corruption, we evaluate the performance of deep ensembles and MultiSWAG varying the training budget. For deep ensembles we show performance as a function of independently trained models in the ensemble. For MultiSWAG we show performance as a function of independent SWAG approximations that we construct; we then sample 20 models from each of these approximations to construct the final ensemble.

While the training time for MultiSWAG is the same as for deep ensembles, at test time MultiSWAG is more expensive, as the corresponding ensemble consists of a larger number of models. To account for situations when test time is constrained, we also propose MultiSWA, a method that ensembles independently trained SWA solutions [15]. SWA solutions are the means of the corresponding Gaussian SWAG approximations. Izmailov et al. [15] argue that SWA solutions approximate the local ensembles represented by SWAG with a single model.

In Figure 4 we show the negative log-likelihood as a function of the number of independently trained models for a Preactivation ResNet-20 on CIFAR-10 corrupted with Gaussian blur with varying levels of intensity (increasing from left to right) in Figure 4. MultiSWAG outperforms deep ensembles significantly on highly corrupted data. For lower levels of corruption, MultiSWAG works particularly well when only a small number of independently trained models are available. We note that MultiSWA also outperforms deep ensembles, and has the same computational requirements at training and test time as deep ensembles. We present results for other types of corruption in Appendix Figures 9, 10, 11, 12, showing similar trends. There is an extensive evaluation of MultiSWAG in the Appendix.

Our perspective of generalization is deeply connected with Bayesian marginalization. In order to best realize the benefits of marginalization in deep learning, we need to consider as many hypotheses as possible through multimodal posterior approximations, such as MultiSWAG. In Section 7 we return to MultiSWAG, showing how it can alleviate double descent, and lead to striking improvements in generalization over SGD and single basin marginalization, for both accuracy and NLL.

## 5 Neural Network Priors

A prior over parameters $p(w)$ combines with the functional form of a model $f(x; w)$ to induce a distribution over functions $p(f(x; w))$. It is this distribution over functions that controls the generalization properties of the model; the prior over parameters, in isolation, has no meaning. Neural networks are imbued with structural properties that provide good inductive biases, such as translation equivariance, hierarchical representations, and sparsity. In the sense of Figure 2, the prior will have large support, due to the flexibility of neural networks, but its inductive biases provide the most mass to datasets which are representative of problem settings where neural networks are often applied. In this section, we study the properties of the induced distribution over functions. We directly continue the discussion of priors in Section 6, with a focus on examining the noisy CIFAR results in Zhang et al. [51], from a probabilistic perspective of generalization. These sections are best read together. In [49] we discuss tempering in connection with these results.

### 5.1 Deep Image Prior and Random Network Features

Two recent results provide strong evidence that vague Gaussian priors over parameters, when combined with a neural network architecture, induce a distribution over functions with useful inductive biases. In the *deep image prior*, Ulyanov et al. [46] show that *randomly initialized* convolutional neural networks *without training* provide excellent performance for image denoising, super-resolution, and inpainting. This result demonstrates the ability for a sample function from a random prior over neural networks $p(f(x; w))$ to capture low-level image statistics, before any training. Similarly, Zhang et al. [51] shows that pre-processing CIFAR-10 with a *randomly initialized*

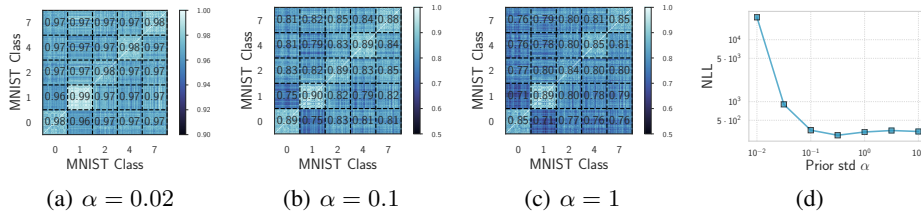

(a) $\alpha = 0.02$      (b) $\alpha = 0.1$      (c) $\alpha = 1$      (d)

Figure 5: **Induced prior correlation function.** Average pairwise prior correlations for pairs of objects in classes $\{0, 1, 2, 4, 7\}$ of MNIST induced by LeNet-5 for $p(f(x; w))$ when $p(w) = \mathcal{N}(0, \alpha^2 I)$. Images in the same class have higher prior correlations than images from different classes, suggesting that $p(f(x; w))$ has desirable inductive biases. The correlations slightly decrease with increases in $\alpha$. **(d)**: NLL of an ensemble of 20 SWAG samples on MNIST as a function of $\alpha$ using a LeNet-5.

*untrained* convolutional neural network dramatically improves the test performance of a simple Gaussian kernel on pixels from 54% accuracy to 71%. Adding $\ell_2$ regularization only improves the accuracy by an additional 2%. These results again indicate that *broad* Gaussian priors over parameters induce reasonable priors over networks, with a minor additional gain from decreasing the variance of the prior in parameter space, which corresponds to $\ell_2$ regularization.

## 5.2 Prior Class Correlations

In Figure 5 we study the prior correlations in the outputs of the LeNet-5 convolutional network [24] on objects of different MNIST classes. We sample networks with weights $p(w) = \mathcal{N}(0, \alpha^2 I)$, and compute the values of logits corresponding to the first class for all pairs of images and compute correlations of these logits. For all levels of $\alpha$ the correlations between objects corresponding to the same class are consistently higher than the correlation between objects of different classes, showing that the network induces a reasonable prior similarity metric over these images. Additionally, we observe that the prior correlations somewhat decrease as we increase $\alpha$, showing that bounding the norm of the weights has some minor utility, in accordance with Section 5.1. Similarly, in panel (d) we see that the NLL significantly decreases as $\alpha$ increases in $[0, 0.5]$, and then slightly increases, but is relatively constant thereafter.

## 6 Rethinking Generalization

Zhang et al. [51] demonstrated that deep neural networks have sufficient capacity to fit randomized labels on popular image classification tasks, and suggest this result requires re-thinking generalization to understand deep learning.

We argue, however, that this behaviour is not puzzling from a probabilistic perspective, is not unique to neural networks, and cannot be used as evidence against Bayesian neural networks (BNNs) with vague parameter priors. Fundamentally, the resolution is the view presented in the introduction: from a probabilistic perspective, generalization is at least a *two-dimensional* concept, related to support (flexibility), which should be as large as possible, supporting even noisy solutions, and inductive biases that represent relative prior probabilities of solutions.

Indeed, we demonstrate that the behaviour in Zhang et al. [51] that was treated as mysterious and specific to neural networks can be exactly reproduced by Gaussian processes (GPs). Gaussian processes are an ideal choice for this experiment, because they are popular Bayesian non-parametric models, and they assign a prior directly in function space. Moreover, GPs have remarkable flexibility, providing universal approximation with popular covariance functions such as the RBF kernel. Yet the functions that are a priori *likely* under a GP with an RBF kernel are relatively simple. We describe GPs further in the Appendix, and Rasmussen and Williams [40] provides an extensive introduction.

We start with a simple example to illustrate the ability for a GP with an RBF kernel to easily fit a corrupted dataset, yet generalize well on a non-corrupted dataset, in Figure 6. In Fig 6(a), we have sample functions from a GP prior over functions $p(f(x))$, showing that likely functions under the prior are smooth and well-behaved. In Fig 6(b) we see the GP is able to reasonably fit data from a

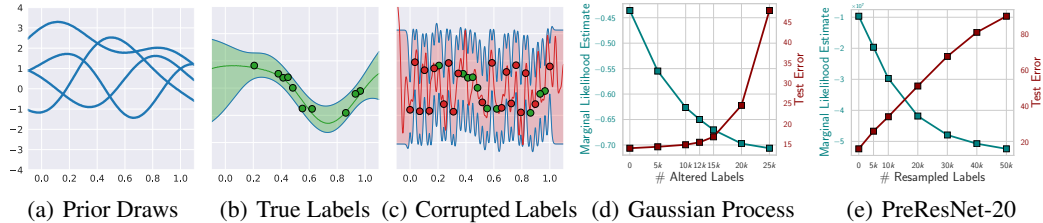

|(a) Prior Draws|(b) True Labels|(c) Corrupted Labels|(d) Gaussian Process|(e) PreResNet-20|

Figure 6: **Rethinking generalization. (a)**: Sample functions from a Gaussian process prior. **(b)**: GP fit (with 95% credible region) to structured data generated as $y_{\text{green}}(x) = \sin(x \cdot 2\pi) + \epsilon,\ \epsilon \sim \mathcal{N}(0, 0.2^2)$. **(c)**: GP fit, with no training error, after a significant addition of corrupted data in red, drawn from Uniform$[0.5, 1]$. **(d)**: Variational GP marginal likelihood with RBF kernel for two classes of CIFAR-10. **(e)**: Laplace BNN marginal likelihood for a PreResNet-20 on CIFAR-10 with different fractions of random labels.

structured function. And in Fig 6(c) the GP is also able to fit highly corrupted data, with essentially no structure; although these data are not a likely draw from the prior, the GP has support for a wide range of solutions, including noise.

We next show that GPs can replicate the generalization behaviour described in Zhang et al. [51] (experimental details in the Appendix). When applied to CIFAR-10 images with random labels, *Gaussian processes achieve 100% train accuracy*, and 10.4% test accuracy (at the level of random guessing). However, the same model trained on the true labels has train and test accuracies of 72.8% and 54.3%. Thus, the generalization behaviour described in Zhang et al. [51] is not unique to neural networks, and can be resolved by separately considering support and inductive biases.

Indeed, although Gaussian processes support CIFAR-10 images with random labels, they are not likely under the GP prior. In Fig 6(d), we compute the approximate GP marginal likelihood on a binary CIFAR-10 classification problem, with labels of varying levels of corruption. We see as the noise in the data increases, the approximate marginal likelihood, and thus the prior support for these data, decreases. In Fig 6(e), we see a similar trend for a Bayesian neural network. Again, as the fraction of corrupted labels increases, the approximate marginal likelihood decreases, showing that the prior over functions given by the Bayesian neural network has less support for these noisy datasets. We provide further experimental details in the Appendix. We provide further remarks on BNN priors, and connections with tempering, in [49].

Dziugaite and Roy [7] and Smith and Le [44] provide complementary perspectives on Zhang et al. [51], for MNIST; Dziugaite and Roy [7] show non-vacuous PAC-Bayes bounds for the noise-free binary MNIST but not noisy MNIST, and Smith and Le [44] show that logistic regression can fit noisy labels on subsampled MNIST, interpreting the results from an Occam factor perspective.

## 7 Double Descent

*Double descent* [e.g., 3] describes generalization error that decreases, increases, and then again decreases, with increases in model flexibility. The first decrease and then increase is referred to as the *classical regime*: models with increasing flexibility are increasingly able to capture structure and perform better, until they begin to overfit. The next regime is referred to as the *modern interpolating regime*, which has been presented as mysterious generalization behaviour in deep learning.

However, our perspective of generalization suggests that performance should monotonically improve as we increase model flexibility when we use Bayesian model averaging with a reasonable prior. Indeed, in the opening example of Figure 1, we would in principle want to use the most flexible possible model. Our results so far show that standard BNN priors induce structured and useful priors in the function space, so we should not expect double descent in Bayesian deep learning models that perform reasonable marginalization.

To test this hypothesis, we evaluate MultiSWAG, SWAG and standard SGD with ResNet-18 models of varying width, following Nakkiran et al. [33], measuring both error and negative log likelihood (NLL). For the details, see Appendix D. We present the results in Figure 7 and Appendix Figure 17.

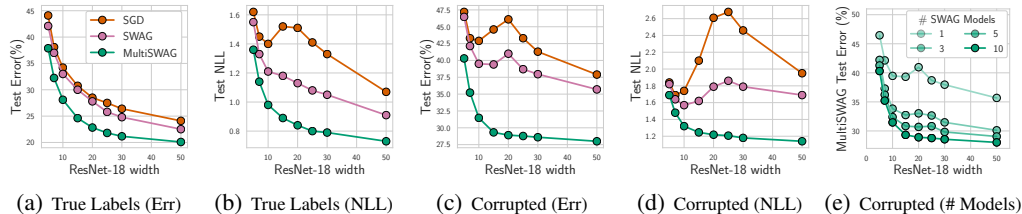

| (a) True Labels (Err) | (b) True Labels (NLL) | (c) Corrupted (Err) | (d) Corrupted (NLL) | (e) Corrupted (# Models) |

Figure 7: **Bayesian model averaging alleviates double descent.** **(a)**: Test error and **(b)**: NLL loss for ResNet-18 with varying width on CIFAR-100 for SGD, SWAG and MultiSWAG. **(c)**: Test error and **(d)**: NLL loss when 20% of the labels are randomly reshuffled. SWAG reduces double descent, and MultiSWAG, which marginalizes over multiple modes, entirely alleviates double descent both on the original labels and under label noise, both in accuracy and NLL. **(e)**: Test errors for MultiSWAG with varying number of independent SWAG models; error monotonically decreases with increased number of independent models, alleviating double descent. MultiSWAG also provides significant performance improvements. See Appendix Figure 17 for additional results.

First, we observe that models trained with SGD indeed suffer from double descent, especially when the train labels are partially corrupted (see panels 7(c), 7(d)). We also see that SWAG, a unimodal posterior approximation, reduces the extent of double descent. Moreover, MultiSWAG, which performs a more exhaustive *multimodal* Bayesian model average *completely mitigates double descent*: the performance of MultiSWAG solutions increases monotonically with the size of the model, showing no double descent even under significant label corruption. We note that deep ensembles follow a similar pattern to MultiSWAG in Figure 7(c), also mitigating double descent, with slightly worse accuracy (about 1-2%). This result is in line with our perspective of Section 3.2 of deep ensembles providing a *better* approximation to the Bayesian predictive distribution than conventional single-basin Bayesian marginalization procedures.

Our results highlight the importance of marginalization over multiple modes of the posterior: under 20% label corruption SWAG clearly suffers from double descent while MultiSWAG does not. In Figure 7(e) we show how the double descent is alleviated with increased number of independent modes marginalized in MultiSWAG. These results also clearly show that MultiSWAG provides significant improvements in *accuracy* over both SGD and SWAG models, in addition to NLL, an often overlooked advantage of Bayesian model averaging.

## 8 Discussion

We have presented a probabilistic perspective of generalization, which depends on the support and inductive biases of the model. The support should be as large possible, but the inductive biases must be well-calibrated to a given problem class. We argue that Bayesian neural networks embody these properties — and through the lens of probabilistic inference, explain generalization behaviour that has previously been viewed as mysterious. Moreover, we argue that Bayesian marginalization is particularly compelling for neural networks, show how deep ensembles provide a practical mechanism for marginalization, and propose a new approach that generalizes deep ensembles to marginalize within basins of attraction. We show that this multimodal approach to Bayesian model averaging, MultiSWAG, can entirely alleviate double descent, to enable monotonic performance improvements with increases in model flexibility, as well significant improvements in generalization accuracy and log likelihood over SGD and single basin marginalization.

There are certainly many challenges to estimating the integral for a Bayesian model average in modern deep learning, including a high-dimensional parameter space, and a complex posterior landscape. But viewing the challenge indeed as an integration problem, rather than an attempt to obtain posterior samples for a simple Monte Carlo approximation, provides opportunities for future progress. Bayesian deep learning has been making fast practical advances, with approaches that now enable better accuracy and calibration over standard training, with minimal overhead.

## Broader Impacts

Improvements in methods and understanding for Bayesian deep learning are crucial for using machine learning in reliable decision making. A well-calibrated predictive *distribution* provides significantly more information for making decisions, and helps protect against rare but costly mistakes in loss-calibrated inference. Bayesian deep learning can also be used for improved sample efficiency, decreasing the need for costly large labelled datasets typically needed to train accurate neural networks. Bayesian neural networks can also be far more robust to noise, as we have shown in the double descent experiments. A better understanding of generalization in deep learning also helps us more reliably predict when a neural network might be reasonable to deploy in real problems. Potential broader drawbacks include increased computation, and increased complexity of the approaches — sometimes requiring expert knowledge on approximate inference to achieve good performance.

### Acknowledgements

This research is supported by an Amazon Research Award, Facebook Research, Amazon Machine Learning Research Award, NSF I-DISRE 193471, NIH R01 DA048764-01A1, NSF IIS-1910266, and NSF 1922658 NRT-HDR: FUTURE Foundations, Translation, and Responsibility for Data Science.

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
