[Supplementary Material]

## Appendix Outline

This appendix is organized as follows. In Section A, we provide background material on Gaussian processes. In Section B, we discuss different ways of approximating the Bayesian model average. In Section C, we present further results comparing MultiSWAG and MultiSWA to Deep Ensembles under data distribution shift on CIFAR-10. In Section D, we provide the details of all experiments presented in the paper.

## A   Gaussian processes

With a Bayesian neural network, a distribution over parameters $p(w)$ induces a distribution over functions $p(f(x; w))$ when combined with the functional form of the network. Gaussian processes (GPs) are often used to instead *directly* specify a distribution over functions.

A Gaussian process is a distribution over functions, $f(x) \sim \mathcal{GP}(m, k)$, such that any collection of function values, queried at any finite set of inputs $x_1, \ldots, x_n$, has a joint Gaussian distribution:

$$f(x_1), \ldots, f(x_n) \sim \mathcal{N}(\mu, K). \tag{2}$$

The mean vector, $\mu_i = \mathbb{E}[f(x_i)] = m(x_i)$, and covariance matrix, $K_{ij} = \mathrm{cov}(f(x_i), f(x_j)) = k(x_i, x_j)$, are determined by the *mean function* $m$ and *covariance function* (or *kernel*) $k$ of the Gaussian process.

The popular RBF kernel has the form

$$k(x_i, x_j) = \exp\left(-\frac{1}{2\ell^2}\|x_i - x_j\|^2\right). \tag{3}$$

The *length-scale* hyperparameter $\ell$ controls the extent of correlations between function values. If $\ell$ is large, sample functions from a GP prior are simple and slowly varying with inputs $x$.

Gaussian processes with RBF kernels (as well as many other standard kernels) assign positive density to any set of observations. Moreover, these models are *universal approximators* [40]: as the number of observations increase, they are able to approximate any function to arbitrary precision.

Work on Gaussian processes in machine learning was triggered by the observation that Bayesian neural networks become Gaussian processes with particular kernel functions as the number of hidden units approaches infinity [35]. This result resembles recent work on the neural tangent kernel [e.g., 16].

## B   Approximating the BMA

In Figure 8, we illustrate the conceptual difference between deep ensembles, a standard variational single basin approach, and MultiSWAG. In the top panel, we have a conceptualization of a multimodal posterior. VI approximates the posterior with multiple samples within a single basin. But we see in the middle panel that the conditional predictive distribution $p(y|x, w)$ does not vary significantly within the basin, and thus each additional sample contributes minimally to computing the marginal predictive distribution $p(y|x, \mathcal{D})$. On the other hand, $p(y|x, w)$ varies significantly between basins, and thus each point mass for deep ensembles contributes significantly to the marginal predictive distribution. By sampling within the basins, MultiSWAG provides additional contributions to the predictive distribution. In the bottom panel, we have the gain in approximating the predictive distribution when adding a point mass to the representation of the posterior, as a function of its location, assuming we have already sampled the mode in dark green. Including samples from different modes provides significant gain over continuing to sample from the same mode, and including weights in wide basins provide relatively more gain than the narrow ones.

## C   Deep Ensembles and MultiSWAG Under Distribution Shift

In Figures 9, 10, 11, 12 we show the negative log-likelihood for Deep Ensembles, MultiSWA and MultiSWAG using PreResNet-20 on CIFAR-10 with various corruptions as a function of independently

Figure 8: **Approximating the BMA.** $p(y|x, \mathcal{D}) = \int p(y|x, w)p(w|\mathcal{D})dw$. **Top:** $p(w|\mathcal{D})$, with representations from VI (orange) deep ensembles (blue), MultiSWAG (red). **Middle:** $p(y|x, w)$ as a function of $w$ for a test input $x$. This function does not vary much within modes, but changes significantly between modes. **Bottom:** Distance between the true predictive distribution and the approximation, as a function of representing a posterior at an additional point $w$, assuming we have sampled the mode in dark green. There is more to be gained by exploring new basins, than continuing to explore the same basin.

(a) Gaussian Noise

(b) Impulse Noise

(c) Shot Noise

Figure 9: **Noise Corruptions.** Negative log likelihood on CIFAR-10 with a PreResNet-20 for Deep Ensembles, MultiSWAG and MultiSWA as a function of the number of independently trained models for different types of corruption and corruption intensity (increasing from left to right).

trained models (SGD solutions, SWA solutions or SWAG models, respectively). For MultiSWAG, we generate 20 samples from each independent SWAG model. Typically MultiSWA and MultiSWAG significantly outperform Deep Ensembles when a small number of independent models is used, or when the level of corruption is high.

In Figure 14, following Ovadia et al. [38], we show the distribution of negative log likelihood, accuracy and expected calibration error as we vary the type of corruption. We use a fixed training time budget: 10 independently trained models for every method. For MultiSWAG we ensemble 20 samples from each of the 10 SWAG approximations. MultiSWAG particularly achieves better NLL than the other two methods, and MultiSWA outperforms Deep Ensembles; the difference is especially

(a) Defocus Blur

(b) Glass Blur

(c) Motion Blur

(d) Zoom Blur

(e) Gaussian Blur

Figure 10: **Blur Corruptions.** Negative log likelihood on CIFAR-10 with a PreResNet-20 for Deep Ensembles, MultiSWAG and MultiSWA as a function of the number of independently trained models for different types of corruption and corruption intensity (increasing from left to right).

pronounced for higher levels of corruption. In terms of ECE, MultiSWAG again outperforms the other two methods for higher corruption intensities.

We note that Ovadia et al. [38] found Deep Ensembles to be a very strong baseline for prediction quality and calibration under distribution shift. For this reason, we focus on Deep Ensembles in our comparisons.

# D   Details of Experiments

In this section we provide additional details of the experiments presented in the paper.

(a) Contrast

(b) Saturate

(c) Elastic Transform

(d) Pixelate

(e) JPEG Compression

Figure 11: **Digital Corruptions.** Negative log likelihood on CIFAR-10 with a PreResNet-20 for Deep Ensembles, MultiSWAG and MultiSWA as a function of the number of independently trained models for different types of corruption and corruption intensity (increasing from left to right).

## D.1 Approximating the True Predictive Distribution

For the results presented in Figure 3 we used a network with 3 hidden layers of size 10 each. The network takes two inputs: $x$ and $x^2$. We pass both $x$ and $x^2$ as input to ensure that the network can represent a broader class of functions. The network outputs a single number $y = f(x)$.

To generate data for the plots, we used a randomly-initialized neural network of the same architecture described above. We sampled the weights from an isotropic Gaussian with variance $0.1^2$ and added isotropic Gaussian noise with variance $0.1^2$ to the outputs:

$$y = f(x; w) + \epsilon(x),$$

with $w \sim \mathcal{N}(0, 0.1^2 \cdot I)$, $\epsilon(x) \sim \mathcal{N}(0, 0.1^2 \cdot I)$. The training set consists of 120 points shown in Figure 3.

For estimating the ground truth we ran 10 chains of Hamiltonian Monte Carlo (HMC) using the `hamiltorch` package [6]. We initialized each chain with a network pre-trained with SGD for 3000

(a) Snow

(b) Fog

(c) Brightness

Figure 12: **Weather Corruptions.** Negative log likelihood on CIFAR-10 with a PreResNet-20 for Deep Ensembles, MultiSWAG and MultiSWA as a function of the number of independently trained models for different types of corruption and corruption intensity (increasing from left to right).

(a) HMC chain 1     (b) HMC chain 2     (c) sample log-probs     (d) $w_i$ marginal

Figure 13: **HMC convergence diagnostics. (a), (b)**: predictive distributions approximated from two different chains of HMC.The predictive distributions are virtually identical despite using different initializations for the chains. **(c)**: Convergence of sample log-probability as a function of HMC iteration. The log-probability fluctuates around a fixed value throughout the sampling and doesn't have a clear trend. **(d)**: Marginal distribution of the weight $w_i$ from a middle layer in the network estimated from samples from a single chain. The marginal distribution resembles a zero-mean Gaussian corresponding to the prior distribution. The presented diagnostics suggest that each of the HMC chains is converged and provides a good coverage of the posterior.

steps, then ran Hamiltonian Monte Carlo (HMC) for $5 \cdot 10^5$ steps at 1000 leapfrog steps per sample, producing 500 samples.

For Deep Ensembles, we independently trained 50 networks with SGD for 20000 steps each. We used minus posterior log-density as the training loss. For SVI, we used a fully-factorized Gaussian approximation initialized at an SGD solution trained for 20000 steps. For HMC and SVI we set prior variance to 0.1 and noise variance to 0.0005. For deep ensembles we had to use a high prior variance of 100 to avoid converging to degenerate solutions.

**HMC covergence diagnostics.** To ensure convergence of our Hamiltonian Monte Carlo sampler we apply several diagnostics. First we look at the difference in predictions between different chains. In Figure 13 (a), (b) we visualize the predictive distributions for two different chains, and they are virtually indistinguishable. To verify this visual intuition, we compute the Gelman–Rubin convergence diagnostic [11] for the predictive distributions at each position in the input space. The diagnostic $\hat{R}$ is

Figure 14: Negative log likelihood, accuracy and expected calibration error distribution on CIFAR-10 with a PreResNet-20 for Deep Ensembles, MultiSWAG and MultiSWA as a function of the corruption intensity. Following Ovadia et al. [38] we summarize the results for different types of corruption with a boxplot. For each method, we use 10 independently trained models, and for MultiSWAG we sample 20 networks from each model. As in Figures 5, 11-14, there are substantial differences between these three methods, which are hard to see due to the vertical scale on this plot. MultiSWAG particularly outperforms Deep Ensembles and MultiSWA in terms of NLL and ECE for higher corruption intensities.

defined as

$$B = \frac{N}{M-1} \sum_{m=1}^{M} (\bar{y}_m - \bar{y})^2, \quad W = \frac{1}{M} \sum_{m=1}^{M} s_m^2, \quad \hat{R} = \sqrt{\frac{\frac{N-1}{N}W + \frac{1}{N}B}{W}}, \tag{4}$$

where $\bar{y}_m$ is the average prediction at a given position $x$ within the chain $m$, $\bar{y}$ is the average prediction at that position across all the chains, $s_m^2 = \frac{1}{N-1} \sum_{i=1}^{N} (y_{mi} - \bar{y}_m)$ is an estimate of the variance of the prediction at position $x$ within the chain $m$, $M$ is the number of chains and $N$ is the length of each chain. Intuitively, the diagnostic $\hat{R}$ measures the ratio of the within-chain and between-chain variances of predictions at a given position. The values of the diagnostic are $1. \pm 0.01$, which is very close to $1$, as desired. Additionally, in Figure 13 in panel (c) we show the convergence of the log-probability of the HMC samples as a function of iteration, and in panel (d) we show the marginal distribution of one of the weights estimated from a single chain. Both diagnostics suggest that each of the chains has converged and provide a good approximation of the predictive distribution.

**Discrepancy with true BMA.** For the results presented in panel (d) of Figure 3 we computed Wasserstein distance between the predictive distribution approximated with HMC and the predictive distribution for Deep Ensembles and SVI. We used the one-dimensional Wasserstein distance function[1] from the *scipy* package. We computed the Wasserstein distance between marginal distributions at each input location, and averaged the results over the input locations. In the top sub-panels of panels (b), (c) of Figure 3 we additionally visualize the marginal Wasserstein distance between the HMC predictive distribution and Deep Ensembles and SVI predictive distrbutions respectively for each input location.

## D.2  Deep Ensembles and MultiSWAG

We evaluate Deep Ensembles, MultiSWA and MultiSWAG under distribution shift in Section 4. Following Ovadia et al. [38], we use a PreResNet-20 network and the CIFAR-10 dataset with different types of corruptions introduced in Hendrycks and Dietterich [14]. For training individual

Figure 15: **Prior correlations under corruption.** Prior correlations between predictions (logits) for PreResNet-20, Linear Model and RBF kernel on original and corrupted images as a function of corruption intensity for different types of corruptions. The lengthscale of the RBF kernell is calibrated to produce similar correlations to PreResNet on uncorrupted datapoints. We report the mean correlation values over 100 different images and show the $1\sigma$ error bars with shaded regions. For all corruptions except Snow, Saturate, Fog and Brightness the correlations decay slower for PreResNet compared to baselines.

SGD, SWA and SWAG models we use the hyper-parameters used for PreResNet-164 in Maddox et al. [29]. For each SWAG model we sample 20 networks and ensemble them. So, Deep Ensembles, MultiSWA and MultiSWAG are all evaluated under the same training budget; Deep Ensembles and MultiSWA also use the same test-time budget.

For producing the corrupted data we used the code[2] released by Hendrycks and Dietterich [14]. We had issues producing the data for the *frost* corruption type, so we omit it in our evaluation, and include *Gaussian blur* which was not included in the evaluation of Hendrycks and Dietterich [14].

### D.3 Rethinking Generalization

In Section 6, we experiment with Bayesian neural networks and Gaussian processes on CIFAR-10 with noisy labels, inspired by the results in Zhang et al. [51] that suggest we need to re-think generalization to understand deep learning.

Following Zhang et al. [51], we train PreResNet-20 on CIFAR-10 with different fractions of random labels. To ensure that the networks fits the train data, we turn off weight decay and data augmentation, and use a lower initial learning rate of 0.01. Otherwise, we follow the hyper-parameters that were used with PreResNet-164 in Maddox et al. [29]. We use diagonal Laplace approximation to compute

Figure 16: **(a)**–**(c)**: Average pairwise prior correlations for pairs of objects in classes $\{0, 1, 2, 4, 7\}$ of MNIST induced by LeNet-5 for $p(f(x; w))$ when $p(w) = \mathcal{N}(0, \alpha^2 I)$. Images in the same class have higher prior correlations than images from different classes, suggesting that $p(f(x; w))$ has desirable inductive biases. The correlations slightly decrease with increases in $\alpha$. Panels **(e)**–**(g)** show sample functions from LeNet-5 along the direction connecting a pair of MNIST images of $0$ and $1$ digits. The complexity of the samples increases with $\alpha$. **(d)**: NLL and **(h)** classification error of an ensemble of 20 SWAG samples on MNIST as a function of $\alpha$ using a LeNet-5. The NLL is high for overly small $\alpha$ and near-optimal for larger values with an optimum near $\alpha = 0.3$.

an estimate of marginal likelihood for each level of label corruption. Following Ritter et al. [42] we use the diagonal of the Fisher information matrix rather than the Hessian.

We perform a similar experiment with a Gaussian process with RBF kernel on the binary classification problem for two classes of CIFAR-10. We use variational inference to fit the model, and we use the variational evidence lower bound to approximate the marginal likelihood. We use variational inference to overcome the non-Gaussian likelihood and not for scalability reasons; i.e., we are not using inducing inputs. We use the `GPyTorch` package [10] to train the models. We use an RBF kernel with default initialization from `GPyTorch` and divide the inputs by $5000$ to get an appropriate input scale. We train the model on a binary classification problem between classes $0$ and $1$.

For the 10-class GP classification experiment we train 10 one-vs-all models that classify between a given class and the rest of the data. To reduce computation, in training we subsample the data not belonging to the given class to $10k$ datapoints, so each model is trained on a total of $15k$ datapoints. We then combine the 10 models into a single multi-class model: an observation is attributed to the class that corresponds to the one-vs-all model with the highest confidence. We use the same hyper-parameters as in the binary classification experiments.

### D.4 Double Descent

In Section 7 we evaluate SGD, SWAG and MultiSWAG for models of varying width. Following Nakkiran et al. [33] we use ResNet-18 on CIFAR-100; we consider original labels, $10\%$ and $20\%$ label corruption. For networks of every width we reuse the hyper-paramerers used for PreResNet-164 in Maddox et al. [29]. For original labels and $10\%$ label corruption we use 5 independently trained SWAG models with MultiSWAG, and for $20\%$ label corruption we use 10 models; for $20\%$ label corruption we also show performance varying the number of independent models in Figures 7(e) and 17(c). Both for SWAG and MultiSWAG we use an ensemble of 20 sampled models from each of the SWAG solutions; for example, for MultiSWAG with 10 independent SWAG solutions, we use an ensemble of 200 networks.

(a) 10% Corrupted (Err)  (b) 10% Corrupted (NLL)  (c) 20% Corrupted (# Models)

Figure 17: **Double Descent. (a)**: Test error and **(b)**: NLL loss for ResNet-18 with varying width on CIFAR-100 for SGD, SWAG and MultiSWAG when 10% of the labels are randomly reshuffled. MultiSWAG alleviates double descent both on the original labels and under label noise, both in accuracy and NLL. **(e)**: Test NLLs for MultiSWAG with varying number of independent models under 20% label corruption; NLL monotonically decreases with increased number of independent models, alleviating double descent.

## Footnotes

[1]https://docs.scipy.org/doc/scipy/reference/generated/scipy.stats.wasserstein_distance.html

[2]https://github.com/hendrycks/robustness/blob/master/ImageNet-C/create_c/make_cifar_c.py