[Reviews · NeurIPS 2020]

Review 1

Summary and Contributions: This paper provides a mix between discussing high-level conceptual ideas and perspectives and presenting a variety of experimental results, all under the umbrella of generalization in (Bayesian) deep learning. More concretely, the central argument of the paper is that Bayesian learning should be primarily viewed as aiming to marginalize over different plausible hypotheses of the data, intead of relying on a single hypothesis (which is what ordinary deep learning is doing). The ultimate goal is thus to accurately estimate the posterior _predictive_ distribution (over outputs), rather than to accurately approximate the posterior distribution (over weights). They thus recommend that Bayesian methods should ideally focus their efforts on carefully representing the posterior distribution in regions that contribute most to the predictive distribution. In this line of thought, they further argue that deep ensembles, one of the state-of-the-art approaches for obtaining well-calibrated predictive distributions, do effectively approximate the Bayesian model average (even if the individual ensemble members are not actually samples from the posterior), and thus should not be considered in competition to Bayesian methods. Another central idea described is that when viewed probabilistically, generalization is mostly determined by two properties of the model: its support (i.e. the set of datasets which it can in theory generate) and its inductive biases (i.e. the relative prior probabilities of actually generating those datasets). This view e.g. offers an explanation to the mysteriously-viewed phenomenon that deep neural networks have enough capacity to fit random noise, yet are successfully able to generalize in many practical tasks. Empirically, the paper investigates a wide variety of phenomena, for example: 1) how marginalizing across different distinct modes of the posterior (as done in deep ensembles) and additionally within those modes (as done in MultiSWAG, which combines deep ensembles with the recently proposed SWAG method for Bayesian deep learning) is competitive/superior to other Bayesian approaches in terms of accuracy, calibration and robustness to distributional shift, 2) how Gaussian processes also exhibit the previously-mentioned phenomenon of both being able to fit random noise as well as generalizing well, 3) how MultiSWAG alleviates the double descent phenomenon in deep neural networks, 4) how uninformative prior distributions over model parameters can result in sensible prior distributions over funcions when using model classes with appropriate/desirable inductive biases, 5) how tempering the posterior interacts with modeling choices to help obtain better predictions.

Strengths: Overall, this is a great paper, which offers the reader a multitude of novel interesting ideas and perspectives, and intriguing empirical studies. This described perspectives are sensible and will help provide orientation for future research in Bayesian deep learning. The experiments are sound, comprehensive and insightful, shedding some light on poorly-understood, important phenomena in deep learning. This work makes significant contributions which are highly relevant to the NeurIPS community and will be of great interest to researchers and practitioners, both working on deep learning and probabilitic modeling.

Weaknesses: One weakness impacting novelty is that the paper does not make any significant methodological contributions. While the extension of SWAG to capture multiple basins of attraction (termed MultiSWAG) is interesting, the idea is rather incremental and straightforward. In fact, the method had already been proposed and assessed in Maddox et al. 2019 (though not under the name MultiSWAG), where it was also shown to be superior to both SWAG and deep ensembles individually (section 5.2 in Maddox et al. 2019: "we can similarly ensemble independently trained SWAG models; an ensemble of 3 SWAG models achieves NLL of 0.6178"). The contribution of this paper is thus not to propose MultiSWAG, but rather to extend its empirical evaluation to further demonstrate its efficacy in comparison to previous approaches, which is done convincingly. Another issue impacting novelty is that some of the main conceptual ideas described in this paper had already been presented in Wilson 2020. That being said, Wilson 2020 appears to be an unpublished tech report that is only available on arXiv, and I am not sure how that report relates to this paper (e.g. if the arXiv report is an earlier version of this paper), so this might not be a significant issue. Furthermore, the storyline of the paper is somewhat incoherent, and there does not appear to be a main line of argument. I.e., the paper does not read as one coherent piece of work, but instead as a collection of related but still somewhat independent ideas and experiments (which are all interesting in themselves, but would each not be significant enough for a paper on their own) that the authors tried to put together in a single conference paper. Overall, I am not sure if the way this content is presented in this conference paper format does the work justice. In particular, this paper comes with a large appendix with further, arguably not less interesting/relevant/significant results and discussions. For example, in the Introduction, the authors specifically mention that several results (e.g. how priors over weights induce priors over functions, and how tempering affects Bayesian deep learning) are only presented in the appendix. In the Discussion, the authors also refer to the appendix for further remarks and discussions (i.e., "about future developments for Bayesian neural network priors, and approaches to research in Bayesian deep learning"), which are insightful and should ideally be part of the main text. All in all, given the large amount of interesting and relevant content, this makes me feel like a journal might be a more appropriate venue for this work. However, I will leave this consideration to the area chair.

Correctness: Yes, the claims and methodology appear correct, as far as I can tell.

Clarity: The paper is very well written overall and thus fairly easy to understand. I really like the introduction, as it intuitively and gently guides the reader into the core points of discussion of this paper and provides an easy-to-understand running example. Figure 2 looks great and nicely illustrates the relationship between a model's support and inductive biases.

Relation to Prior Work: The related work section provides a fairly thorough overview over the literature in Bayesian neural networks, and the paper provides pointers to relevant work throughout. For the results shown in Figure 3 (c) (and referenced in l. 192), the authors might want to cite Foong et al 2019, which empirically showed that Bayesian neural networks trained with mean-field variational inference underestimate uncertainty (and thus yield overconfident predictions) in-between seperated clusters of data. I am not fully sure how this work relates to Wilson 2020 ("The case for Bayesian deep learning"), which does contain substantial overlap in content in terms of the high-level discussions and perspectives, but is only briefly mentioned in the related work section. It would be great if the authors could clarify this relationship, which is important for the assessment of novelty (not necessarily to the reviewers, as this would break double blindness, but e.g. to the area chair).

Reproducibility: Yes

Additional Feedback: POST-REBUTTAL: Thank you for commenting on some of the concerns and questions I had raised. After carefully reading the other reviews and taking into account your rebuttal, I am unfortunately less enthusiastic about this paper than initially. In particular, reviewer #2 pointed out some important issues with the experimental evaluation, and after another close look at the experiments, I agree that some of them are less significant than I initially thought. Given that the methodological contribution is low and that the paper is entirely empirical, the execution of the experiments is certainly critical. The internal reviewer discussion also made me realise that the issues I had raised on the many additional results presented in the appendix might be more significant, and I now have a stronger feeling that a journal might be more appropriate if the authors wish to present all these results in a single manuscript, in order to allow all those results to be scrutinised closely. (To clarify: This comment is not intended to judge the quality of those results in any way -- in fact, those results look very promising at first sight, but given that they're presented in the appendix, the reviewers cannot be expected to carefully check them, which would be important if they were to be published.) Overall, while I still appreciate the merits of the paper and am thus still leaning towards acceptance due to the potential impact of the paper, I will not fight to have the paper accepted anymore. I believe that this paper investigates important open questions and will thus be a significant contribution to our understanding of BNNs, but only when the experimental issues have been resolved and there is a better solution for the additional experiments in the appendix. As a result, I am lowering my score from 8 to 6. ============== Questions: - l. 138: what do you mean by "often"? in which cases do we want to compute that predictive distribution, and in which cases do we not? - you seem to argue that the true predictive distribution is the holy grail that all methods should aim to approximate; are there any theoretical results showing that the "true" predictive distribution is superior to every other predictive distribution in practically relevant metrics (e.g. quantifying accuracy, calibration, robustness), etc.? - Figure 3: it is interesting to consider the (average) Wasserstein distance to the "true" predictive distribution as a quality metric for methods that approximate the predictive distribution, which I haven't seen before; why did you use the Wasserstein distance to measure the discrepancy between predictive distributions? have you considered alternative measures, such as other integral probability metrics (e.g. MMD), or alpha-divergences (e.g. KL divergence)? did you observe qualititve differences? - do the metrics we commonly use to assess BDL methods implicitly capture (or correlate with) the discrepancy to the true predictive distribution? do you think that we should be using such metrics, and do you have ideas for how such metrics would look like (in cases when the true predictive is intractable)? - it appears like you are only considering uniform weights for the Gaussian mixture approximation resulting from MultiSWAG; would it be beneficial to estimate/learn non-uniform mixture weights? - Figure 6: how do deep ensemble perform here? does double descent occur for them or not? Minor issues: - l. 20: the claim "most audiences overwhelmingly favour choices (1) and (2), for fear of overfitting" is not substantiated; did you perform an empirical study to verify this claim? it would be interesting to see if this is actual the case - all Figures: the font sizes should be larger (for both the axis labels and units) to aid readability - Figure 2: are the colors you used color-blind friendly? - l. 152: the acronym MCMC is not introduced - l. 299: it is not fully clear what it means for inductive biases to be "well-calibrated to a given problem class"; I understand what the authors are trying to say, but perhaps there is a better way of expressing this (especially given that the notion of "calibration" already has a distinct technical meaning)


Review 2

Summary and Contributions: The paper provides high level background material on Bayesian inference in the context of deep discriminative models. It argues that deep ensembles can be viewed naturally through this lens. This motivates generalisations of ensembles of distributional approximations and the paper applies this idea to the SWAG approximations. It then considers such methods in the context of a simple toy regression experiment, out-of-distribution generalisation, the rethinking generalisation experiment, and double descent experiments.

Strengths: A well written paper on an interesting topic. The paper contains a basic almost tutorial like introduction to Bayesian ideas in the context of neural networks that could be useful for deep learning experts who want to move into this area.

Weaknesses: The technical contribution is very simple. This isn’t a bad thing per se, but it does raise the bar on the quality and significance of the experimental evaluation. My view is that the current experimental contribution needs to be substantially tightened up before this bar is reached. ------- ----- Update after reading the other reviews, author response, and reviewer discussion (put here due to character limits in the other entry boxes): I think the paper is border line for the following reasons: The technical contribution — of ensembling over multiple basins — is modest in my opinion. Two of the main experiments are interesting, two don’t bring much to the table currently. If the two that are interesting could be tightened up, this would make the paper significantly stronger and the take homes clear. Specifically: Experiment 1 (shown in Figure 3) does not appear rigorous to me. The authors have not addressed whether HMC was validated as being close to ground truth which is key for substantiating their claims. The response does not address this. Experiment 2: This experiment shows that multiSWAG provides an advantage over deep ensembles for small ensemble sizes in terms of held out log likelihood on out-of-distribution data, but that the gap is quite close when there are 10 members of the ensemble for some of the noise corruptions. I would like more detail on these experiments to understand the significance of these results. The authors responded to my suggestion that "it would also be useful to report accuracy / held out NLL on the original task test set”. However, it’s my understanding that Supplementary Fig. 18 does not show this (as the authors’ response claims). I was after the uncorrupted accuracies / held out log-likelihoods, whereas this shows the values for the corrupted OOD sets. In [57] upon which Fig 18 is based, there is an additional first ’test’ column showing these on the standard test set. However, perhaps these test set values are close to the values obtained by corruption 1? It would be very useful to see the accuracy counterparts to figs 14-17. This would help understand whether there are trade-offs between accuracy and calibration (often the case for in distribution data). Experiment 3 on 'Rethinking Generalization’: As laid out in my original review, I don’t think this brings much to the table. Experiment 4. I think it is interesting that the double descent behaviour is not apparent for ensembles. It’s also good that the authors have said that they will add deep ensembles to this experiment and that they have run one such experiment. In the response they say "In the setting of Fig. 6 (c), deep 12 ensembles achieve errors 60., 65., 68.6, 70., 70.4, 70.3, 71.2, 71.4 for widths 5, 7, 10, 15, 20, 25, 30, 50 respectively." I believe these are accuracies rather than errors. At least by eye, these numbers are quite close to the multiSWAG results — perhaps a small improvement of about 1% at the larger widths? It’s hard to assess the efficacy of multiSAG without seeing full results for all conditions here. In terms of the bigger picture, another key question still remains to my mind — is all this additional compute better spent on training a single larger model with SGD than a smaller model with SWAG / multiSWAG? For these reasons, I have not changed my score.

Correctness: Yes.

Clarity: The paper is well written.

Relation to Prior Work: Yes, but see the detailed review for complementary prior work on ensembles of BNNs.

Reproducibility: Yes

Additional Feedback: Personally I found that the exposition focussed a lot on quite basic background concepts in the probabilistic modelling and probabilistic inference methodology, but then was short on necessary technical details about the methods and experiments. Specifically, I would like to have seen more background information on: SWAG and especially multiSWAG should be carefully explained in the main text with equations / pseudo-code (I couldn’t find these in the paper or the supplementary material) discussions about computational overhead (memory and CPU time) at training and test times of the compared methods (Deep Ensembles, SWAG, mutliSWAG, SWA, and VI) More experimental details (see later) I think that a tutorial on Bayesian methods for deep learning would be a valuable contribution, but in my view this can’t come at the expense of explaining the methods and experiments and as NeurIPS papers are so short, this is probably not the right place for the tutorial like elements. I have several specific comments about experiment 1 which is shown in Figure 3: "We provide details for generating the data and training the models in Appendix I.1.” This looks to be Appendix G.1. Why were the fitted models here given a mismatched prior distribution over the weights (the standard deviation is 100 times larger than the weights in the generative process of the underlying model) and a mismatched observations noise (five times smaller in the trained model than in the data generative process). Do the findings hinge on this? "A close approximation of the true predictive distribution obtained by combining 200 HMC chains” and "we can near-exactly compute the predictive distribution” These claims needs backing up — that the approximation is close — since it is notoriously difficult to get provably reliable samples from the posterior even in simple looking cases like this. Indeed, an adversarial reviewer might say 'the method used for finding ground truth (initialise HMC chains with SGD) is very similar to DeepEnsembles so it is not surprising that these two are found to be the most similar’. Is HMC changing the weight values by a large amount? Do the sample chains appear to be converging? Do experiments on simple toy examples (e.g. the GP limit or linear networks) show the approximate inference works on these unit tests? To be clear, I am not a fan of VI, but I don’t find this experiment is rigorous enough to prove much. In many ways the natural comparison to VI — in terms of the amount of compute that is being used and the fact that multiple different means / variances are being learned — would be ensembles of VI. These have been shown to perform very well in comparison to deep ensembles before: Neural network ensembles and variational inference revisited, Marcin B Tomczak, Siddharth Swaroop, Richard E Turner, 1st Symposium on Advances in Approximate Bayesian Inference, 2018 In this context, I find figure 3d a little misleading as it is not a like-for-like comparison. Experiment 2 in figure 4 was simple but interesting. It would be very useful to include plots of the accuracy as a function of the number of models (either in the main text or in the supplementary material). E.g. are all the models making very poor predictions under distribution shift, but multiSWAG knows that it is uncertain and has quite uniform output probabilities? Or is the accuracy of multiSWAG also much better on the out of distribution examples? It would also be useful to report accuracy / held out NLL on the original task test set to demonstrate that we’re not trading in-domain performance for out-of-domain performance. I don’t have much to say on the 'Rethinking Generalization’ section except that I agree with the authors’ view here. However, as researchers like Pirmin Lemberger and Thomas Dietterich have pointed out (see their open review comments on the paper’s thread), I think that this should have been clear from the start no matter whether you take a probabilistic modelling or another perspective. The experiments on the Double Descent phenomena was more interesting. The accuracy improvements over SGD are strong, but of course this comes with a large increase in compute. It is not clear to me that this compute might be better spent on training a single larger model with SGD than a smaller model with SWAG. DeepEnsembles was also missing as a key baseline in this experiment. With these omissions, I again have concerns about the significance of the conclusions that can be drawn from this experiment. Minor comments "The key distinguishing property of a Bayesian approach is marginalization, rather than using a single setting of weights.” I disagree with this statement — there are lots of ways to average models. It’s not useful to claim that they are all Bayesian. I think section J of the supplementary material provides a more nuanced perspective than this that I largely agree with. For example, it says "Broadly speaking, what makes Bayesian approaches distinctive is a posterior weighted marginalization over parameters.” Here the marginalisation is w.r.t. a distribution that has a natural interpretation as an approximate posterior which is more restrictive than the statements in the main text. Line 146 "We then discuss moving beyond Simple Monte Carlo…” -> “In the next section we then discuss moving beyond Simple Monte Carlo” As phrased currently it’s not clear what’s in the main paper and what is in the supplementary material. In the related work I’d mention: An Ensemble of Bayesian Neural Networks for Exoplanetary Atmospheric Retrieval, Adam D. Cobb et al The Astronomical Journal, 2019 Supplementary: Line 1039 "Just like we how expectation propagation [51]"


Review 3

Summary and Contributions: The paper addresses the marginalization process needed for computing the predictive posterior in Bayesian deep networks. The paper examines deep ensembles as a method for obtaining an effective mechanism for approximate Bayesian marginalization. Starting from this result, the authors investigate the prior over functions implied by a vague distribution over neural network weights, and show that a vague prior does not undermine generalization, contrary to recent speculation in the field.

Strengths: The idea of marginalizing within basins of attraction is novel and interesting. It is important that it is empirically shown to yield predictive densities with much more robust properties, without significant computational overhead. Another important result of the paper is that it provides results showing that Bayesian model averaging alleviates double descent. This is an important result that may inspire further research in the field of Bayesian deep nets; especially important is the result that double descent is avoided even under significant label corruption.

Weaknesses: Following the discussion with other reviewers, and the associated rebuttal, I do share the opinion that the content included in the supplementary (which, I admit, I barely examined in the review phase) introduces findings and statements that should have been discussed in the main manuscript, and appropriately scrutinised. This makes me change my mind as to whether the manuscript is ready for publication at this instance.

Correctness: The paper experimental setup and derivations are correct.

Clarity: The paper is well written

Relation to Prior Work: The prior work review is satisfactory.

Reproducibility: Yes

Additional Feedback: I have read the rebuttal. It clarified some issues, but also left some points that need further consideration.


Review 4

Summary and Contributions: The paper discusses generalisation from a perspective of Bayesian model averaging and touches on a lot of phenomena and topics that have recently been investigated within the wider Deep Learning literature. The paper offers a new perspective of the relation between deep ensembles and Bayesian Deep Learning and argues that deep ensembles also perform Bayesian Model Averaging. Further, it tackles the capability of deep learning to fit random data and shows that GPs have the same capabilities. Lastly, it applies Stochastic Weight Averaging and SWAG to deep ensembles and shows how MultiSWA(G) can be used to overcome the double descent phenomenon.

Strengths: The paper is very well written and easy to follow. It offers interesting insights into recent topics of discussion within the wider field of Deep Learning. The authors show an extensive set of experiments validating the usefulness of Bayesian Model Averaging as well as providing insightful take-aways about the behaviour of deep learning models. This paper could build an important foundation for future research into a better understanding of the behaviour of neural networks and build a bridge between strict 'Bayesians' and deep learning practitioners by framing deep ensembles as a Bayesian method.

Weaknesses: Even though the paper is well written, simple to follow and provides an extensive set of experiments. However, the paper is relatively dense and a lot of further insight can be found in the appendix (which could almost already make another paper). Nevertheless, certain aspects of the analysis seem to be brushed over or left out: - Section 4 starts with the statement that it has been shown that deep ensembles indeed perform Bayesian Model Averaging, however no part of Section 3 directly addresses this point. It is clear that ensembles perform some sort of model averaging, however it would be interesting to further discuss the form q(w|D) takes in deep ensembles and which implicit assumptions about the prior are made and whether deep ensembles recover the true p(w|D) in the limit of infinite models in the ensemble (Sampling perspective?) or whether they are closer to a variational approximation and which implicit assumptions are made about the form of q(w|D). - Fig 3 offers a comparison of different predictive distributions and it is claimed that an increase in samples does not improve the estimate of the predictive distribution for SVI. However, I believe that this is an unfair comparison as this estimate is strongly influenced by the form of variational approximation used. Therefore, an approximation with little expressive power can only gain little improvements by further samples, whereas more complicated variational distributions might cover multiple modes in the weight distribution and could benefit from more samples. This should me more thoroughly examined.

Correctness: The claims and methods are correct as far as I can judge.

Clarity: The paper is very well written and easy to follow. It was a pleasure to read. The figures are of very high quality.

Relation to Prior Work: Previous works has been well addressed and compared to.

Reproducibility: Yes

Additional Feedback: Did you test the behaviour of regular ensembles on the double descent phenomenon? Does regular ensembling also alleviate this behaviour?

[Author Response · NeurIPS 2020]

We thank the reviewers for their thoughtful comments and support. We want to emphasize that the paper provides many timely insights unified by a strong narrative around probabilistic model construction and generalization, showing the role of multimodal marginalization, neural networks priors, tempering, support, and inductive biases — with significant demonstrations, including an exhaustive empirical study of marginalization, a demonstration of the role of marginal likelihood in resolving questions around generalization, and a Bayesian resolution of double descent (tying together the opening narrative and the importance of multimodal marginalization). The material is particularly timely, given recent questions about Bayesian methods in deep learning, such as the treatment of deep ensembles as a competing approach to Bayesian neural networks, and the cold posterior experiments in Wenzel et. al (ICML 2020), which are resolved in our discussion of tempering. While papers with many types of contributions can be difficult to assess, we believe this paper makes an important and timely contribution to NeurIPS, and appreciate the strong support of reviewers.

Inspired by reviewer comments, we additionally evaluated the effect of deep ensembles. In the setting of Fig. 6 (c), deep ensembles achieve errors $60., 65., 68.6, 70., 70.4, 70.3, 71.2, 71.4$ for widths $5, 7, 10, 15, 20, 25, 30, 50$ respectively. In agreement with our Bayesian perspective of deep ensembles, they almost resolve double descent and provide similar but worse results compared to MultiSWAG. We will add the results to Figure 6.

- **R1.** Thank you for the thoughtful and supportive review. We describe the unifying narrative and contributions above. We appreciate the feedback and we will further clarify these connections in a final version. As you say, the main contribution of the MultiSWAG part of the paper is not the algorithm but the Bayesian perspective, the exhaustive empirical study, and the resolution of double descent. We believe these are major contributions, and that the paper overall has a significant degree of novelty (which we believe should not be constrained to algorithmic innovation).
- We will follow up with the AC on the paper you mention, but we can assure you this brief note is in no way in conflict with our submission and should not weigh in the decision.
- L138: in some applications such as continual learning we may want to approximate the posterior over the parameters rather than the posterior predictive distribution. The Bayesian posterior predictive is not necessarily the optimal model average under model misspecification. In particular, we argue for temperature scaling in appendix E, which leads to a different predictive distribution compared to the *true* posterior.
- Figure 3: Wasserstein distance can be easily computed from samples from the two distributions and provides a useful measure of their difference without the mode seeking or mode covering behavior of KL divergence.
- The weights of the components in MultiSWAG should reflect the relative mass of the different modes in the posterior. We expect these masses to be similar, but estimating them empirically is an interesting direction for future work.
- **R2.** Thank you for your thoughtful review. We think there are some misunderstandings, and hope you can consider our clarifications in your updated assessment. We would like to emphasize that the material in this paper should not be viewed as a background or tutorial on BDL, leading to the proposal of MultiSWAG. Our paper presents a novel perspective as well as truly exhaustive experiments that provide insights into BDL. The main purpose of MultiSWAG is to demonstrate properties of multi-basin marginalization, as part of the larger narrative of the paper.
- We will add a more detailed description of MultiSWA & MultiSWAG but note that these straightforwardly ensemble independently trained SWAG and SWA models (which are described in full papers) and the point here is about multibasin marginalization. MultiSWAG has the same computational complexity as Deep Ensembles at training time, and the memory overhead comes from storing 5-20 copies of the weights for each mode (note that these weights can be stored on the disk rather than in GPU memory and take significantly less memory compared to the activations). At test time, MultiSWAG has more overhead due to samples within each basin.
- In Fig. 3 the vague prior was used to prevent over-regularization of the network that would lead to a poor fit of the ground truth (obtained with HMC) to the data. The results do not hinge on the specific values of the hyper-parameters.
- We want to clarify that Fig. 3 is not intended to provide a like-for-like comparison. This experiment is intended to show the importance of multibasin representations for approximate BMA integration. In particular, we do not argue against VI in general, and a multibasin ensemble of VI approximations would also support our findings. We will clarify, and include a discussion of the reference you suggested.
- We do actually have accuracy results for Exp 2 in the Appendix, Fig. 18. We also report accuracy for double descent.
- We include deep ensembles to our experiment on double descent as you suggested (see above).
- **R3.** Thank you for your supportive review. We have now included a new experiment on deep ensembles performance in the context of double descent.
- **R4.** Thanks for your supportive review. (1) We view deep ensembles as a compelling mechanism for approximate BMA integration under constraints – different from both variational methods and MCMC, which are typically combined with simple MC. In this vein, we do not view deep ensemble weights as samples from an approximate posterior, and we would not recover the exact predictive distribution by taking an infinite number of ensembles. We also provide a related discussion in Appendix C and Appendix Figure 8. We will clarify in the final version. (2) Re: Fig 3. The point here is more about multimodal vs unimodal in approximating the BMA integral than about a deficiency of variational methods. Multiple independently trained VI models (for a multi-basin posterior) could indeed perform well and also make this point. Thanks for the question. We will clarify.

[Meta-Review · NeurIPS 2020]

After much discussion, the reviewers largely converged towards recommending to accept this submission. The reviewers appreciate the merits of the paper, believe it investigates important open questions, and will thus be a significant contribution to our understanding of BNNs, but only when the experimental issues mentioned in the reviews are resolved. I would draw the author's attention to the fact that the reviewers raised concerns about the supplementary material containing a number of sections which are not connected to results in the main paper (on tempered posteriors, sampling from the prior, discussions of what’s Bayesian, PAC Bayes etc.). Per reviewing guidelines, since these sections were not relevant for understanding the main paper, these were not reviewed with scrutiny. However, the reviewers found strong statements in the unreviewed supplementary material involving other recent work which they believe deserve close scrutiny if they are to be published. They therefore question whether the conference format is appropriate for this work, or whether the paper should instead be presented in a format to allow for closer scrutiny for all presented results (eg journal submission). However, the reviewers did not highlight anything seriously wrong with the supplementary material, therefore this cannot be considered grounds for rejection. The AC would want to stress that NeurIPS is not providing a strong stamp of approval on supplementary material since the reviewers are not strictly responsible to review all such material. Having this paper published is not an explicit sanction of the supplementary material, and the authors are advised to remove unrelated material at the camera ready to avoid potential confusion. Specifically, the reviewers asked to remove the line on a Bayesian perspective on tempering from the abstract as it isn't covered in the main paper, and perhaps to remove the section in the supplementary material as concerns were raised about it: a) the claims given are speculation and are not backed up with experimental or theoretical justification, and b) they involve a very broad definition of what is Bayesian that includes model mismatch or correcting for approximate inference etc. and this is not reflected by the sentence in the abstract.